# Effects of Long-Term Endurance Exercise and Lithium Treatment on Neuroprotective Factors in Hippocampus of Obese Rats

**DOI:** 10.3390/ijerph17093317

**Published:** 2020-05-10

**Authors:** Jusik Park, Wookwang Cheon, Kijin Kim

**Affiliations:** 1Department of Taekwondo, College of Physical Education, Keimyung University, Daegu 42601, Korea; parkjs@kmu.ac.kr; 2Department of Physical Education, College of Physical Education, Keimyung University, Daegu 42601, Korea; wk11106@gw.kmu.ac.kr

**Keywords:** lithium, exercise, obesity, hippocampus

## Abstract

To investigate the effects of long-term lithium treatment and low intensity endurance exercise on brain-derived neurotrophic factor (BDNF) expression and glycogen synthase kinase 3 beta (GSK3β) activity in the hippocampus of obese rats. Fifty 10-week-old male Sprague-Dawley rats were selected. There was a control group of 10 rats (chow control group) while the other forty rats were fed on a high-fat diet for eight weeks to induce obesity. Rats were then assigned into four random groups. The rats were given 10 mg/kg lithium chloride (LiCl) dissolved in 1 mL sterile distilled water once a day, 5 times a week. The rats did 20 min of treadmill walking with an exercise intensity of 40% maximal oxygen uptake (VO_2_ max) (12 m/min, slope 0%). This was performed for 20 min a day, 3 days a week. Twelve weeks of lithium treatment or endurance exercise significantly reduced body weight and body fat mass in obese rats, without showing additive effects when the treatments were given in parallel or significant toxic responses in alanine aminotransferase (ALT) and aspartate aminotransferase (AST) levels in blood and kidney and liver tissues. BDNF expression in the hippocampus was significantly increased both in exercise and lithium groups with synergistic effects found in the group where both exercise and lithium treatments were given in parallel. On the other hand, the decrease in GSK3β activity was shown only in the lithium treatment group, without showing additive effects when the treatments were given in parallel. Lithium and low-intensity endurance exercise for 12 weeks increased the expression of BDNF, a neuroprotective factor in the hippocampus of obese mice. Lithium treatment alone inhibited the activity of GSK3β. This can be interpreted as a positive indication of applicability of the two factors in the prevention of neurodegenerative diseases.

## 1. Introduction

Neurodegenerative disorders collectively refer to destructive diseases characterized by loss of nerve function and viability, leading to deterioration of brain function, including motor, memory and cognitive abilities [1]. As life expectancy increased over the past centuries, the prevalence rate of age-related disorders, such as neurodegenerative diseases, continued to increase [2,3]. Alzheimer’s disease, Parkinson’s disease, Huntington’s disease and multiple sclerosis are chronic disorders characterized by neuronal loss of motor, sensory or cognitive systems. Numerous studies previously conducted have identified a strong link between obesity or metabolic syndrome and neurodegeneration, showing that neurodegenerative disorders are caused by increased insulin resistance and expression of inflammatory factors in the neurons and decreased expression of brain-derived neurotrophic factor (BDNF). Middle age obesity has been reported to be a risk factor for dementia, Parkinson’s disease [4], Alzheimer’s and vascular dementia [5,6] in later life. Therefore, neurodegenerative diseases should be considered a problem for all people in Korea where the obese population has rapidly increased due to a Westernized lifestyle and lack of physical activity, rather than viewing it as a problem limited to aging population and the elderly. However, as most studies focus on the treatment rather than prevention, preventative measures should be found immediately. 

Endurance exercise has been reported to improve neurological disorders such as depression, epilepsy, stroke, Alzheimer’s disease and Parkinson’s disease [7,8,9,10]. One of the mechanisms by which endurance exercise benefits the brain is the induction of neuronal / growth factors, particularly the brain-derived neurotrophic factor (BDNF). BDNF promotes brain development by inducing the survival, differentiation, migration, dendritic branching and synapse formation of neurons [11,12]. In fact, BDNF is also essential for synaptic plasticity, hippocampal function and learning [13]. Therefore, the decrease in BDNF concentration is considered a biological indicator for memory and general cognitive impairment caused by dementia [14]. Increasing the level of stimulation of neuronal cell production through increased BDNF expression can reduce the risk of dementia and play a positive role in cognitive function [15]. However, despite the merits of exercise, most studies that report an increase in BDNF in humans have reported a significant effect when exercising at a moderate intensity [16,17,18,19]. Therefore, adults who do not exercise and especially those who are obese, cannot benefit from the neuroprotective effects of exercise and hence finding an adjuvant therapy or alternative substance is necessary. 

Lithium produces neuroprotective effects and stimulates neurogenesis through multiple signal transduction pathways [20,21,22,23,24,25,26,27]. Previous studies show that lithium suppresses GSK-3 (Glykogensynthasekinase-3) in neurons and increases brain-derived neurotrophic factor (BDNF), B-cell lymphoma-2 (Bcl-2) and heat shock protein 70 (HSP-70), while reducing pro-apoptotic factors [28,29]. In fact, lithium also reduces the level of neuronal death, microglial activation, cyclooxygenase-2 induction, amyloid-β (Aβ) and hyperphosphorylated tau, improving learning ability and memory by preserving the integrity of blood–brain barrier and relieving neurological defects and psychotic disorders [30]. Based on the results of previous studies, endurance exercise and lithium treatments are expected to have a positive effect on obesity-induced neurodegenerative disorder prevention through BDNF activation and GSK3β inhibition in the hippocampus of obese rats.

Therefore, this study compared the effects of long-term endurance exercise or lithium treatment on BDNF expression and GSK3β activity in the hippocampus of obese rats to examine the applicability of lithium as a neuroprotective substance.

## 2. Materials and Methods

### 2.1. Research Subject and Method 

Fifty 10-week-old male Sprague-Dawley rats were procured and subjected to a week-long adaptation period. While 10 rats formed the control group (Chow control group, CC), 40 rats were fed with a high-fat diet for eight weeks to induce obesity then assigned into four random groups, (Fat-diet control group, FC; Lithium group, Li; Exercise group, Ex; Lithium + Exercise group, Lex). Subsequently had, twelve weeks of lithium and endurance exercise treatments. The calorie content of the high-fat diet (DIO Teklad rodent diet, Envigo, UK) consisted of 30% carbohydrate, 50% fat and 20% protein vitamins (22 g/kg Teklad vitamins mix no. 40077), minerals (51 g/kg Teklad mineral mix no. 170915), methionine (5 g/kg, Teklad Premier no. 10850) and choline chloride (1.3 g/kg) [31]. Water and food were freely available to the rats. Rats were housed two per cage (20.7 × 35 × 17 cm) and the temperature and humidity inside the cages were maintained respectively at 21 °C and 50%. Each of the light and dark periods was set to twelve hours, and the body weight and feed intake were measured every two days during the experiment. This study was conducted as approved by the Animal Experimentation Ethics Committee of Daegu Techno Park BioHealth Convergence Center (BHCC-IACUC-2018-02).

### 2.2. Lithium Treatment

According to the results of previous studies, the dose of lithium showing neuroprotective activity is 10–400 mg/kg, which is equivalent to a blood concentration of 0.6–0.75 mmol/L. Neuroprotective activity among rats was also found at a similar-to-human blood concentration of 0.3–0.7 mmol/L [20,22,23,24,25,26,27]. Hence, this study partially modified previous studies conducted on animals [32,33] and 10 mg/kg of LiCl (Lithium chloride, L4408, Sigma-Aldrich, St. Louis, MO, USA) dissolved in 1 mL of sterile distilled water was orally administered once a day at the same time (10:00 a.m.). One milliliter of sterile distilled water was also orally administered to other groups to put the same stress caused by oral administration on all groups. 

### 2.3. Exercise Protocol 

Endurance exercise was conducted using an electric laboratory treadmill (Quinton Instrument, Seattle, WA, USA). Since the purpose of this study was to observe the interaction between exercise and lithium, exercise was set to low-intensity (40% VO2max, 12 m/min, slope 0%) walking, at which lactic acid does not accumulate in the blood. This was done for 3 days a week and 20 min per day, based on the findings of Koltai et al. [34] and Tang et al. [35] in order to minimize motor stimulation. Adaptation to the exercise was incrementally carried out over the first week; walking on the treadmill at a speed of 7 m/min (slope 0%) for 5 min on Day 1, walking on the treadmill at a speed of 10 m/min (slope 0%) for 10 min on day and finally reaching the designated exercise intensity of walking (12 m/min, slope 0%) for 20 min on Day 3.

### 2.4. Tissue and Blood Sampling

Rats were allowed to rest for 48 h after 12 weeks of treatment to rule out the last-bout exercise effect. Subsequently, the subjects went on a fast of 12 h before receiving pentobarbital sodium (5 mg/100 g) of anesthetic. Then, 5 mL of blood was obtained from the abdominal artery by opening the abdominal cavity and was treated with 50 μL of heparin to prevent coagulation. Plasma was obtained by centrifuging (1500× *g*, 15 min) the collected blood, and was stored at −80 °C for analysis. After blood sampling, the liver and kidneys were removed from the body and paraffin-fixed for H&E staining. Similarly, the hippocampus was separated from the brain after removing it from the body and was stored at −80 °C for analysis, after freezing it with liquid nitrogen. In addition, the retroperitoneal, epididymal and mesenteric fat pads were separately removed and weighed for body fat measurement.

### 2.5. Analysis

#### 2.5.1. H&E Staining 

To test the toxicity of Lithium, the liver and kidney samples which had been fixed with 10% formalin solution for 2 days, were treated with paraffin, cut into 4 µm thickness and attached to glass slides. Hematoxylin and eosin (H&E) staining was carried out on the attached tissues which went through deparaffinization and hydration for observation. The slide was made into a virtual slide file using Aperios’s Scanscope XT (Aperio, CA92081, San Francisco, CA, USA), and kidney and liver damage was analyzed using Imagescope (Aperio, ver10.2.2.2319, San Francisco, CA, USA).

#### 2.5.2. Blood Factor Analysis 

Plasma aspartate aminotransferase (AST) and alanine aminotransferase (ALT) concentrations were measured using an ELISA kit (Sigma-Aldrich, St. Louis, MO, USA).

#### 2.5.3. Western Blotting

Hippocampus tissue was homogenized by using an ice-cold buffer (250 mM sucrose, 10 mM HEPES/1 mM EDTA (pH 7.4), 1 mM Pefabloc, 1 mM EDTA, 1 mM NaF, 1 g/mL aprotinin, 1 g/mL leupeptin, 1 g/mL pepstatin, 0.1 mM bpV (phen) and 2 mg/mL glycerophosphate) [36]. Three cycles of freezing/thawing were carried out on the homogenized and centrifuged material (700× *g*, 10 min), and the protein was quantified in line with Lowry et al. [37]. The sample was dissolved in Laemmli buffer and dispensed into SDS-polyacrylamide gel for electrophoresis. Immunoblotting was performed using anti-BDNF (ab203573, Abcam, Cambridge, UK), anti-GSK3β (CST 9315, CST Inc, San Francisco, CA, USA), and anti-phospho GSK3β (CST 9336, CST Inc., San Francisco, CA, USA), visualized using ECL and quantified by densitometry (sigma-plot 8.0 system, Systat Software Inc., Chicago, Illinois, USA) after using secondary antibodies.

### 2.6. Statistical Analysis

The results from each measurement were calculated as mean and standard error (Mean ± SE), and statistical analysis was performed using SigmaPlot 12.0 statistical package (Systat Software Inc., Chicago, Illinois, USA). One-way ANOVA was conducted to verify the differences among the groups, and two-way repeated measures ANOVA was performed to analyze the group-by-time interaction. Tukey’s method was used for the post-hoc test and the statistical significance level was set to α = 0.05.

## 3. Results

### 3.1. Body Composition

The high-fat diet groups showed a statistically significant higher amount of daily calorie intake (*p* < 0.05) compared to the chow control group and increased gradually over 12 weeks (Table 1). Interestingly, the Li group showed a slightly higher dietary amount than the other high-fat diet intake groups, including the Ex and Lex group, but there were no statistically significant differences (Table 1).

Body weight increased gradually over 12 weeks in all five groups, and the high-fat diet groups showed a more statistically significant increase in body weight compared to the general diet group (*p* < 0.05) (Table 2). There was no significant difference in body weight among the high-fat diet groups of Li, Ex, and Lex, but the body weight increase of the three groups gradually decreased compared to the FC group, showing a statistically significant decrease at the 12th week (*p* < 0.05). After 12 weeks of treatment, total body fat mass in Li, Ex, and Lex groups was significantly decreased compared to FC group (*p* < 0.05). In particular, there was a significant decrease in the amount of retroperitoneal and mesenteric fat pads in all of the three groups (*p* < 0.05). FC and Li groups did not show a significant difference in epididymal fat pads, but the Ex and Lex groups showed a statistically significant difference (*p* < 0.05, Table 3).

### 3.2. Toxicity Test

To assess the toxicity of lithium treatment, levels of ALT and AST, which are factors of liver damage, were measured and no significant difference was found among different groups (Table 4). In addition, H&E staining performed after removing the liver and kidney showed no damage in the liver and kidney tissues (Figure 1). Despite a significant increase in body fat in the FC group, there was no significant microvesicular steatosis in the liver. Since this study focused on whether or not toxicity is caused by lithium treatment, this part is presented as a limitation that is not considered important.

### 3.3. BDNF Production 

The production level of BDNF protein in the hippocampus was significantly increased in the Li, Ex, and LEx groups compared to the CC group, and that of the Ex, LEx groups was significantly increased compared to the FC group (*p* < 0.05). In particular, the LEx group showed a significantly increased BDNF compared to the other four groups (Figure 2).

### 3.4. GSK3β/Phospho-GSK3β Expression Ratio

To determine the activity of GSK3β in the hippocampus, the membranes were stripped for GSK3β measurement and quantified by comparing each band. The results showed that the GSK3β activity of the FC group was the highest and that of the Ex group was higher than that of the CC group (*p* < 0.05). However, the Li and LEx groups demonstrated a statistically significant decrease in the activity of GSK3β compared to the FC group, showing a similar level with the CC group (*p* < 0.05, Figure 3). 

## 4. Discussion

The purpose of this study is to evaluate effects of obesity-induced neurodegenerative disorder prevention by analyzing the effects of long-term lithium and low intensity endurance exercise on the expression of BDNF and GSK3β in the hippocampus of high-fat diet-induced obese rats. In order to do so, a high-fat diet was used to induce obesity for 8 weeks before 12 weeks of lithium treatment or endurance exercise. Additionally, by keeping the motor stimulus lower than the lactic acid threshold, the masking effect from overly high motor stimulus was eliminated when lithium and exercise treatments were given in parallel. The results showed that BDNF expression in the hippocampus was significantly increased in the exercise groups over the 12 weeks and a synergistic effect was found in the group which was given exercise and lithium treatments in parallel. However, the decrease in the activity of GSK3β was observed only in the lithium treatment group without showing an additional effect for integrated treatments. Lithium inhibits GSK3 activity directly or indirectly by increasing serine phosphorylation of GSK-3α and β isoforms [38,39,40,41,42]. According to a number of previous studies, GSK-3 has a pro-apoptotic function and can lead to mood disorders, schizophrenia, diabetes mellitus and various neurological and neuropathic diseases. Thus, lithium, as a GSK-3 inhibitor, has been considered one of the most important therapeutic drugs to treat such diseases [43,44,45]. Lithium produces a GSK3 inhibitory effect through various mechanisms; it rapidly activates ASK to increase GSK3 serine phosphorylation, thereby blocking glutamate-induced Akt inactivation to primarily protect the brain nerves from glutamate-induced excitotoxicity [46]. In addition, lithium upregulates Bcl-2 and inhibits glutamate-induced p53 and Bax [47]. Induction of BDNF is necessary for the neuroprotective action of lithium to take place and GSK3 inhibition by lithium is reported to activate BDNF promoter IV [48,49]. Chuang et al. [50] found that the lithium-induced promoter IV can be also activated by transfection using GSK3 inhibitors, siRNAs of GSK3α or GSK3β. Similar to previous studies, the results of this study show that 12 weeks of lithium treatment significantly reduces the activity of GSK3β and increases the expression of BDNF. What should be noted is that the low-intensity endurance exercise group showed no significant changes in GSK3β activity while exhibiting increased expression of BDNF and there was a synergistic increase in BDNF expression found in the group to which lithium and endurance exercise treatments were given in parallel. Many studies have shown that endurance exercise has an influence on GSK3β expression in brain tissue [51,52] and Liu et al. [53] demonstrated that long-term swimming significantly increases CREB/BDNF and AKT/GSK3β signaling in the hippocampus of rats as much as in other rats exposed to long-term moderate levels of stress. In addition, forced-swimming tests (FST) on stress-induced rats showed suppressed GSK3 activation which has the same effect as antidepressants. However, no clear mechanism of controlling the expression of exercise-induced GSK3 has yet been established. The reason why endurance exercise did not affect GSK3 activity in this study could be due to various factors such as intensity, amount, duration, time and method of measurement of motor stimulus and differences between study subjects [54]. Most previous studies [55,56] that reported changes in GSK3 β activation by exercise training applied protocols with high exercise intensity or relatively high exercise quantity, and Yang et al. [57] reported that GSK3β showed little change at low strength, but it was reported to be significantly changed at medium and high strength. In addition, Pena et al. [58] suggested that GSK3β hardly changed even when a relatively large amount of exercise was applied by using a combination of aerobic and resistance exercise. In this study, in order to observe the interaction between exercise and lithium, it was considered that the change in exercise-induced GSK3β was not clearly seen when applied with low intensity from the viewpoint of minimizing exercise stimulation.

BDNF is induced in various parts of the brain during endurance exercise and it is exponentially induced in the hippocampus [16,17,18,19,59,60]. Neeper et al. [61] reported a dose-dependent increase in BDNF mRNA level in the hippocampus as a result of granting free access to a running wheel to rats for 2, 4 and 7 days. Similarly, Russo-Neustadt et al. [62] found that 20 days of spontaneous aerobic activity produced an explosive increase in BDNF mRNA in CA1, CA3 and CA4 in the hippocampus and dentate gyrus of rats. Zoladz et al. [63] also revealed that moderate endurance training increases serum and plasma BDNF in humans. Although the mechanism of hippocampal BDNF expression by motor stimulus has not yet been identified [62,64], it has been reported that the activation of BDNF-induced TrkB increases neuronal survival and synaptic plasticity by stimulating mitogen-activated protein kinase (MAPK) and PI3K/Akt signaling pathways [65,66]. In addition, spontaneous exercise has been found to further activate this pathway [59,67]. In particular, the major components of the PI3K/Akt pathway, including GSK-3β and the mammalian target of rapamycin (mTOR), have been reported to act as depression and antidepressants [68]. Similar to the results of previous studies, BDNF levels in the hippocampus were significantly increased in the endurance exercise group, showing an additional increase in the integrated treatment group. It seems that exercise and lithium have a different stimulation mechanism when increasing BDNF levels in the hippocampus of obese rats. However, the results of this study alone cannot confirm this speculation and additional research must be carried out. An exercise-induced potential mechanism in the hippocampus of obese rats could be supported by recent research results showing as improved hippocampal insulin signaling, neuroplasticity [69], alleviation of neuroinflammation and cerebrovascular damage [70].

Potential side effects of lithium use include kidney abnormalities, such as decreased urine concentration, which refer to polyuria with excessive thirst and diabetes insipidus in clinical terms [71,72,73]. In fact, previous studies have shown that remarkable decreases in urine enrichment due to disturbed acid-base homeostasis can cause glomerular dysfunction, leading to kidney failure [74]. Thus, pathological analysis of blood liver damage factors in liver and kidney tissues was conducted but no toxic response was found. In addition, as there was no significant difference in weight gain and dietary intake, the amount of lithium used in this study can be considered safe. 

The limitation of this study is that after obesity treatment by high-fat diet induction, BDNF activation and GSK3β inhibition were confirmed by endurance exercise and lithium treatment, but no direct change in brain function was confirmed. Subsequent studies will require detailed analysis of direct effects on brain function, including the synergistic effects of endurance exercise and lithium treatment.

## 5. Conclusions

Lithium and low-intensity endurance exercise for 12 weeks increased the expression of BDNF, a neuroprotective factor in the hippocampus of obese mice, and lithium treatment inhibited the activity of GSK3β. Therefore, long-term low-intensity endurance exercise, when used in parallel with lithium, can be used to protect the brain nerves of obese rats and as a preventative measure against the development of neurodegenerative disorders in the future.

## Figures and Tables

**Figure 1 ijerph-17-03317-f001:**
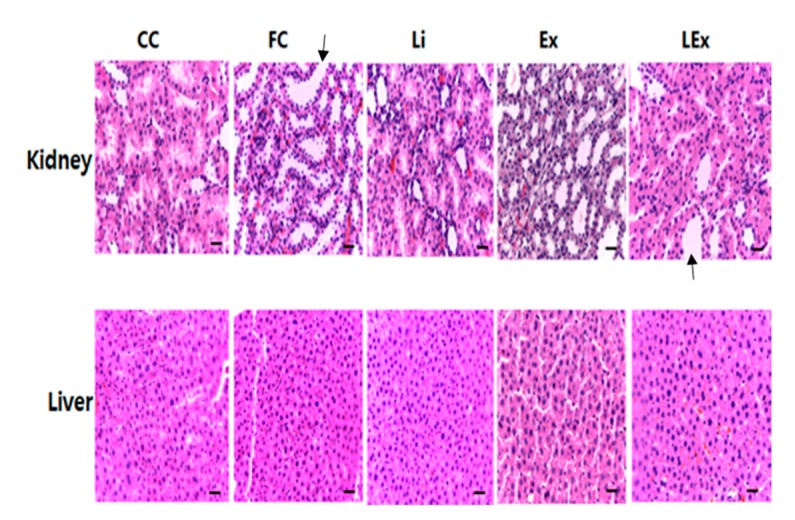
Hematoxylin and eosin staining in kidney and liver tissue. (Histological changes were evaluated in nonconsecutive histological fields, randomly chosen at a magnification of 100×. Scale bar, 100 μm; black arrow indicates a normal state that cannot confirm the degeneration of the tubule in kidney).

**Figure 2 ijerph-17-03317-f002:**
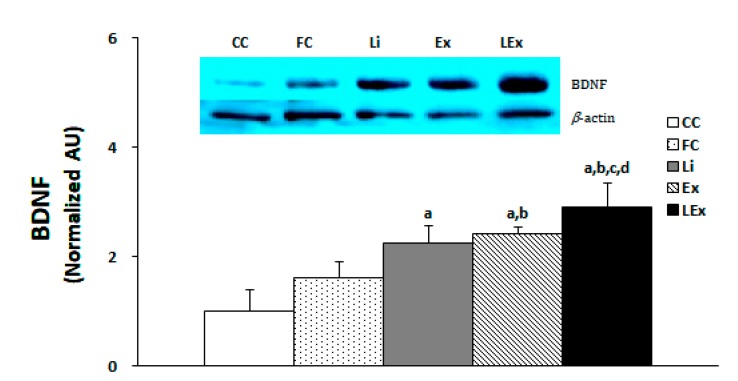
Level of BDNF protein expression. a: Significantly different from CC (*p* < 0.05). b: Significantly different from FC (*p* < 0.05). c: Significantly different from Li (*p* < 0.05). d: Significantly different from Ex (*p* < 0.05).

**Figure 3 ijerph-17-03317-f003:**
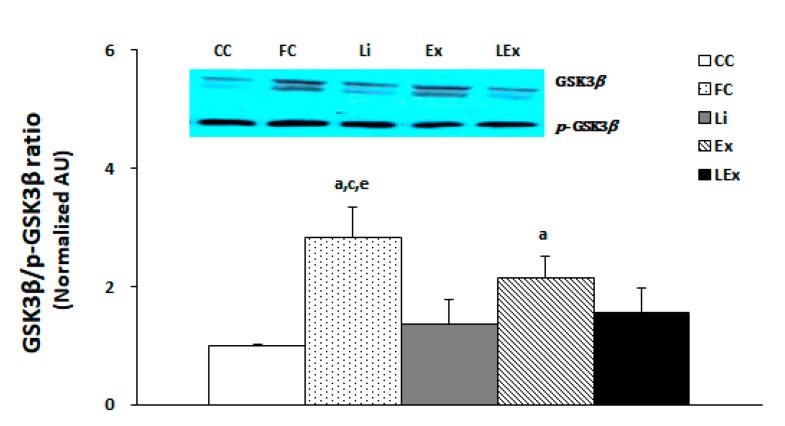
GSK3β/p-GSK3β protein expression ratio. a: Significantly different from CC (*p* < 0.05). c: Significantly different from Li (*p* < 0.05). e: Significantly different from LEx (*p* < 0.05).

**Table 1 ijerph-17-03317-t001:** Daily calorie consumption (kcal/day).

Group	1 Week	3 Week	6 Week	9 Week	12 Week
CC	39.9 ± 4.5	48.7 ± 1.8	52.4 ± 1.8	56.9 ± 3.8	57.5 ± 4.2
FC	53.0 ± 3.3 ^a^	64.0 ± 0.8 ^a^	68.5 ± 2.3 ^a^	73.0 ± 0.8^a^	72.0 ± 0.9 ^a^
Li	50.1 ± 2.6 ^a^	63.0 ± 1.9 ^a^	65.3 ± 4.6 ^a^	82.4 ± 2.1 ^a^	77.3 ± 0.5 ^a^
Ex	49.3 ± 4.4 ^a^	57.3 ± 0.9 ^a^	69.8 ± 2.3 ^a^	67.3 ± 2.4 ^a^	70.9 ± 1.3 ^a^
Lex	50.0 ± 4.4^a^	67.2 ± 4.3^a^	76.0 ± 1.9 ^a^	75.9 ± 0.2 ^a^	72.9 ± 0.3 ^a^

Values are means ± SE. ^a^; Significantly different from CC (*p* < 0.05). CC, Chow control group; FC, Fat-diet control group; Li, Lithium group; Ex, Exercise group; Lex, Lithium + Exercise group.

**Table 2 ijerph-17-03317-t002:** Changes of body weight (g).

Group	0 Week	8 Week	12 Week
CC	301.0 ± 3.0	403.0 ± 13.0	462.0 ± 9.5
FC	300.0 ± 2.7	500.4 ± 1.2 ^a^	590.2 ± 20.0 ^a^
Li	299.8 ± 2.0	503.8 ± 1.5 ^a^	536.5 ± 8.4 ^a,b^
Ex	302.1 ± 1.5	505.0 ± 6.4 ^a^	555.8 ± 16.0 ^a,b^
Lex	300.8 ± 0.9	502.0 ± 3.5 ^a^	527.3 ± 24.2 ^a,b^

Values are means ± SE, ^a^; Significantly different from CC (*p* < 0.05), ^b^; Significantly different from FC (*p* < 0.05).

**Table 3 ijerph-17-03317-t003:** Comparisons of fat fad mass after 12 weeks (g).

Group	Retroperitoneal	Epididymal	Mesenteric	Visceral
CC	16.39 ± 1.63	15.88 ± 0.14	10.97 ± 1.12	43.24 ± 5.02
FC	28.46 ± 0.72 ^a^	23.12 ± 0.66 ^a^	18.64 ± 1.12 ^a^	70.23 ± 1.02 ^a^
Li	23.00 ± 2.00 ^a,b^	20.15 ± 1.63 ^a^	14.65 ± 0.97 ^a,b^	56.04 ± 3.01 ^a,b^
Ex	22.05 ± 2.61 ^a,b^	15.42 ± 0.86 ^b,c^	14.13 ± 1.23 ^a,b^	51.61 ± 4.35 ^a,b^
Lex	24.75 ± 1.50 ^a,b^	16.64 ± 0.80 ^b^	15.89 ± 1.33 ^a^	59.49 ± 4.10 ^a,b^

Values are means ± SE, ^a^; Significantly different from CC (*p* < 0.05), ^b^; Significantly different from FC (*p* < 0.05).

**Table 4 ijerph-17-03317-t004:** Blood level of ALT and AST after 12 weeks (U/L).

	CC	FC	Li	Ex	Lex
ALT	2.58 ± 3.0	2.53 ± 0.43	2.11 ± 0.21	2.92 ± 0.42	2.15 ± 0.39
AST	0.35 ± 0.02	0.39 ± 0.02	0.36 ± 0.01	0.37 ± 0.02	0.35 ± 0.02

Values are means ± SE. ALT, alanine aminotransferase; AST, aspartate aminotransferase.

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
