# Peer review of "Effects of Long-Term Endurance Exercise and Lithium Treatment on Neuroprotective Factors in Hippocampus of Obese Rats"

_ijerph, 2020, doi:10.3390/ijerph17093317_

Round 1

Reviewer 1 Report

The study was well conducted and well written. Despite the simplicity of the methodological procedures, the results are interesting since it demonstrates that the combination of Lithium and low-intense exercise enhance hippocampal BDNF while decreases GSK3beta in diet-induced obese rats. 

Author Response

Response to Reviewer 1 Comments

The study was well conducted and well written. Despite the simplicity of the methodological procedures, the results are interesting since it demonstrates that the combination of Lithium and low-intense exercise enhance hippocampal BDNF while decreases GSK3beta in diet-induced obese rats. 

Response : Thank you. We don't think there is any special point.

Reviewer 2 Report

Twelve weeks of lithium and low-intensity exercise showed a significant increase in BDNF expression in the hippocampus synergistically in contrast to no synergistic decrease in GSK3β in high-fat induced obese SD rats. Authors interpreted these results as a positive indication of applicability of the two factors in the prevention of neurodegenerative diseases.

This conclusion remarks are partially acceptable with regard to BDNF. However, results about GSK3β are not appropriately discussed or concluded, which is the most severe flaw in this manuscript. Simply speaking, GSK3β is not relevant to the synergistic neuroprotective effects of lithium and low-intensity exercise in obese model, and authors should discuss this point more clearly and precisely, not neglecting the discussion using expressions as ‘additional experiments are needed’ or ‘additional research must be carried out’.

There are some minor points to be addressed.

  1. Conclusion remarks in the abstract: ‘GSK3β, a neuroprotective factor in hippocampus in obese rats’ is misleading expression. Also, conclusion remarks ‘Integrated treatment of low-intensity endurance exercise and lithium reduced BDNF expression and GSK3β activity’ is wrong (page 8).
  2. Page 2, line 20-23: ‘Nonetheless, despite many benefits from exercise, most of the studies found an increase in BDNF in humans reported that such effects are produced only when stimulated by moderate or higher-intensity exercise [16-18].’
  3. This sentence is too long and redundant, making the meaning unclear.
  4. Page 2, 1st paragraph of 2.1. Research Subject and Method: This protocol is quite complicated and hard to understand. This part should be described as Figure.     5g / kg, Teklad Premier no.10850: What is this component?
  5. Please indicate the company of high-fat diet.
  6. Page 3, 2.4. Tissue and Blood Sampling: ‘the test subjects were subjected to’ ‘the blood sample was extracted’ These English are strange and should be rephrased.
  7. Page 3, 2.5.1. HE staining: ‘the diameter of the muscle fiber was measured’ I think no muscle fiber in the liver or the kidney.
  8. Page 5, line 4: Table 3 instead of Table 2 at the end of this sentence. There is no description of table 2 in the text.
  9. Regarding Table 2 in itself, there is no significant difference indicated by symbol ‘c’.
  10. Page 5: No subheading of 3.2. Toxicity Test
  11. Second paragraph: Table 4 instead of Table 3.
  12. Page 6: 3.3. BDNF Expression: The term of ‘expression’ should be used for mRNA. In case of protein, please use ‘amount’ or ‘production’.
  13. Figure 2 and 3: P<0.05 instead of P>0.05.

Author Response

Response to Reviewer 2 Comments

Point 1: Twelve weeks of lithium and low-intensity exercise showed a significant increase in BDNF expression in the hippocampus synergistically in contrast to no synergistic decrease in GSK3β in high-fat induced obese SD rats. Authors interpreted these results as a positive indication of applicability of the two factors in the prevention of neurodegenerative diseases. This conclusion remarks are partially acceptable with regard to BDNF. However, results about GSK3β are not appropriately discussed or concluded, which is the most severe flaw in this manuscript. Simply speaking, GSK3β is not relevant to the synergistic neuroprotective effects of lithium and low-intensity exercise in obese model, and authors should discuss this point more clearly and precisely, not neglecting the discussion using expressions as ‘additional experiments are needed’ or ‘additional research must be carried out’.

Response 1: Yes, it was thought of as an appropriate intellectual and was supplemented by adding the following in the discussion.

It is difficult to draw a clear conclusion only from the results of this study, and additional experiments are needed.

  • Most of the previous studies [55,56] that reported changes in GSK3 β activation by exercise training applied protocols with high exercise intensity or relatively high exercise quantity, and Yang et al. [57] reported that GSK3β showed little change at low strength, but it was reported to be significantly changed at medium and high strength. In addition, Pena et al. [58] suggested that GSK3β hardly changed even when a relatively large amount of exercise was applied by applying a combination of aerobic and resistance exercise. In this study, in order to observe the interaction between exercise and lithium, it was considered that the change in exercise-induced GSK3β was not clearly seen when applied with low intensity from the viewpoint of minimizing exercise stimulation.

There are some minor points to be addressed.

Point 2 : Conclusion remarks in the abstract: ‘GSK3β, a neuroprotective factor in hippocampus in obese rats’ is misleading expression. Also, conclusion remarks ‘Integrated treatment of low-intensity endurance exercise and lithium reduced BDNF expression and GSK3β activity’ is wrong (page 8).

Response 2: It was revised as follows.

In abstract :

Twelve weeks of lithium and low-intensity endurance exercise inhibited the activity of GSK3β, a neuroprotective factor in hippocampus in obese rats, while increasing the expression of BDNF.

  • Lithium and low-intensity endurance exercise for 12 weeks increased the expression of BDNF, a neuroprotective factor in hippocampus in obese mice, and lithium treatment inhibited the activity of GSK3β.

In Conclusion :

Integrated treatment of low-intensity endurance exercise and lithium reduced BDNF expression and GSK3β activity’

  • Lithium and low-intensity endurance exercise for 12 weeks increased the expression of BDNF, a neuroprotective factor in hippocampus in obese mice, and lithium treatment inhibited the activity of GSK3β.
  •  

Point 3 : Page 2, line 20-23: ‘Nonetheless, despite many benefits from exercise, most of the studies found an increase in BDNF in humans reported that such effects are produced only when stimulated by moderate or higher-intensity exercise [16-18]. This sentence is too long and redundant, making the meaning unclear.

Response 3: It was revised as follows.

Nonetheless, despite many benefits from exercise, most of the studies found an increase in BDNF in humans reported that such effects are produced only when stimulated by moderate or higher-intensity exercise [16-18].

  • However, despite the merit of exercise, most of the studies that report an increase in BDNF in humans have reported a significant effect when exercising over moderate intensity [16-18].

Point 4 : Page 2, 1st paragraph of 2.1. Research Subject and Method: This protocol is quite complicated and hard to understand. This part should be described as Figure.5g / kg, Teklad Premier no.10850: What is this component?

Response 4: It was revised as follows.

High-fat diet consisted of 30% carbohydrate, 50% fat and 20% protein of the total calories mixed with vitamins (22g / kg Teklad vitamins mix no. 40077), minerals (51g / kg Teklad mineral mix no. 170915), 5g / kg, Teklad Premier no.10850) and choline chloride (1.3g / kg) was used [31].

  • High-fat diet consisted of 30% carbohydrate, 50% fat and 20% protein of the total calories mixed with vitamins (22g / kg Teklad vitamins mix no. 40077), minerals (51g / kg Teklad mineral mix no. 170915), methionine (5g / kg, Teklad Premier no.10850) and choline chloride (1.3g / kg) was used [31].

Point 5 : Please indicate the company of high-fat diet.

Response 5: It was revised as follows.

P.2 L84-85 High-fat diet

-> High-fat diet (DIO Teklad rodent diet, Envigo, UK)

Point 6 : Page 3, 2.4. Tissue and Blood Sampling: ‘the test subjects were subjected to’ ‘the blood sample was extracted’ These English are strange and should be rephrased.

Response 6: It was revised as follows.

After twelve weeks of treatment, the test subjects were subjected to 48 hours of rest to eliminate last-bout exercise effects.

  • Rats were allowed to rest for 48 hours after 12 weeks of treatment to rule out the last-bout exercise effect.
  •  

Plasma from the blood sample was extracted by centrifugation (1500 g, 15 minutes)

  • Plasma was obtained by centrifuging (1500 g, 15 minutes) the collected blood,

Point 7 : Page 3, 2.5.1. HE staining: ‘the diameter of the muscle fiber was measured’ I think no muscle fiber in the liver or the kidney.

Response 7: It was revised as follows.

and the diameter of the muscle fiber was measured using the Imagescope (Aperio technologies, ver10.2.2.2319).

  • and kidney and liver damage were analyzed using the Imagescope (Aperio technologies, ver10.2.2.2319).
  •  

Point 8 : Page 5, line 4: Table 3 instead of Table 2 at the end of this sentence. There is no description of table 2 in the text.

Response 8: It was revised as follows.

Table 2 -> Table 3

Point 9 : Regarding Table 2 in itself, there is no significant difference indicated by symbol ‘c’.

Response 9: It was revised as follows.

c; Significantly different from Li (p<.05) : remove

Point 10: Page 5: No subheading of 3.2. Toxicity Test

Response 10: It was added as follows.

3.2. Toxicity Test

Point 11 : Second paragraph: Table 4 instead of Table 3.

Response 11: It was revised as follows.

Table 3 -> Table 4

Point 12: Page 6: 3.3. BDNF Expression: The term of ‘expression’ should be used for mRNA. In case of protein, please use ‘amount’ or ‘production’.

Response 12: It was revised as follows.

‘3.3. BDNF Production’ or ‘The production level of BDNF’

Point 13 : Figure 2 and 3: P<0.05 instead of P>0.05.

Response 13: It was revised as follows.

  • p<0.05 in Figure 2 and 3

Reviewer 3 Report

In this study, authors present interesting data about the effects of long-term endurance exercise and lithium treatment on neuroprotective factors in hippocampus of obese rats. The author proposed that the obesity could influence on the level of brain-derived neurotrophic factor (BDNF) and causing the neurodegeneration. Then, the study was to determine the neuroprotective factors in the obese rat model. The manuscript is well written, has important message about the effects of exercise and lithium treatment, however the results are not well presented and would probably need to be revised.  

Major concerns:

  1. It is very interesting that the daily calorie consumption of the lithium treatment group, Li was higher than exercise group, FC and Lithium + Exercise group at 9 and 12 week.
  2. The lithium treatment rat has no toxicity in liver and kidney after high fat-diet feeding for 12 weeks. The high fat diet increased body weight and fat fad mass in SD-rat. (Table 3). But in the Fig 1, the tissue staining of liver in fat-diet control group, microvesicular steatosis could not be found.
  3. Several studies have shown that lithium chloride is an inhibitor of the GSK3β. The lithium inhibits the activity of GSK3β through lowering the phosphorylation level. Fig 3 showed that lithium decreased the expression of the GSK3β but not the phosphorylation level.

Minor concerns:

  1. In page 2, line 89: “two rats….” should be corrected.
  2. In page 4, line 146: Could you add the catalog no. of the anti-phospho GSK3β (Molecular Probes).
  3. In page 8, line 284: In the conclusion session, “Integrated treatment of low-intensity endurance exercise and lithium reduced BDNF expression”…….. Actually, the fig 3 these treatment increased the BDNF expression. It should be check again.
  4. In the result session, the 3.4. Author determine the activity of GSK3β in hippocampus. But, the fig 3 was shown the change in the expression level of GSK3β not protein kinase activity. It could correct the description in the paper.

Author Response

Response to Reviewer 3 Comments

In this study, authors present interesting data about the effects of long-term endurance exercise and lithium treatment on neuroprotective factors in hippocampus of obese rats. The author proposed that the obesity could influence on the level of brain-derived neurotrophic factor (BDNF) and causing the neurodegeneration. Then, the study was to determine the neuroprotective factors in the obese rat model. The manuscript is well written, has important message about the effects of exercise and lithium treatment, however the results are not well presented and would probably need to be revised.  

Major concerns:

Point 1: It is very interesting that the daily calorie consumption of the lithium treatment group, Li was higher than exercise group, FC and Lithium + Exercise group at 9 and 12 week.

Response 1: Yes, it is. However, there was no statistically significant difference. However, for clarity, it was modified as follows.

The high-fat diet groups showed statistically significant higher amount of daily calorie intake (p <05) compared to the general diet group and increased gradually over 12 weeks. However, there was no difference in daily calorie intake among the high-fat diet groups (Table 1).

  • The high-fat diet groups showed statistically significant higher amount of daily calorie intake (p <05) compared to the chow control group and increased gradually over 12 weeks. Interestingly, Li group showed a slightly higher dietary amount than the other high-fat diet intake groups, including the Ex and LEx group, but there was no statistically significant differences (Table 1).

Point 2: The lithium treatment rat has no toxicity in liver and kidney after high fat-diet feeding for 12 weeks. The high fat diet increased body weight and fat fad mass in SD-rat. (Table 3). But in the Fig 1, the tissue staining of liver in fat-diet control group, microvesicular steatosis could not be found.

Response 2 : It was considered as an important intellectual and the following sentence was added to the result description.

  • Despite significant increase in body fat in the FC group, there was no significant microvesicular steatosis in the liver. Since this study focused on whether or not toxicity is caused by lithium treatment, this part is presented as a limitation that is not considered important.

Point 3: Several studies have shown that lithium chloride is an inhibitor of the GSK3β. The lithium inhibits the activity of GSK3β through lowering the phosphorylation level. Fig 3 showed that lithium decreased the expression of the GSK3β but not the phosphorylation level.

Response 3 : It was considered as an important intellectual and the following sentence was added to the discussion.

‘Most of the previous studies [55,56] that reported changes in GSK3 β activation by exercise training applied protocols with high exercise intensity or relatively high exercise quantity, and Yang et al. [57] reported that GSK3β showed little change at low strength, but it was reported to be significantly changed at medium and high strength. In addition, Pena et al. [58] suggested that GSK3β hardly changed even when a relatively large amount of exercise was applied by applying a combination of aerobic and resistance exercise. In this study, in order to observe the interaction between exercise and lithium, it was considered that the change in exercise-induced GSK3β was not clearly seen when applied with low intensity from the viewpoint of minimizing exercise stimulation.’

Minor concerns:

Point 4: In page 2, line 89: “two rats….” should be corrected.

Response 4 : It was revised as follows.

two rats were kept in a cage (20.7×35×17cm)

  • Rats were housed two per cage (20.7 x 35 x 17 cm)

Point 5: In page 4, line 146: Could you add the catalog no. of the anti-phospho GSK3β (Molecular Probes).

Response 5 : It was revised as follows.

Immunoblotting was performed using anti-BDNF (Abcam), anti-GSK3β, and anti-phospho GSK3β (Molecular Probes),

  • Immunoblotting was performed using anti-BDNF (ab203573, Abcam, UK), anti-GSK3β (CST 9315, CST Inc, USA), and anti-phospho GSK3β (CST 9336, CST Inc, USA),

Point 6: In page 8, line 284: In the conclusion session, “Integrated treatment of low-intensity endurance exercise and lithium reduced BDNF expression”…….. Actually, the fig 3 these treatment increased the BDNF expression. It should be check again.

Response 6: It was revised as follows.

In Conclusion:

Integrated treatment of low-intensity endurance exercise and lithium reduced BDNF expression and GSK3β activity’

  • Lithium and low-intensity endurance exercise for 12 weeks increased the expression of BDNF, a neuroprotective factor in hippocampus in obese mice, and lithium treatment inhibited the activity of GSK3β.
  •  

Point 7: In the result session, the 3.4. Author determine the activity of GSK3β in hippocampus. But, the fig 3 was shown the change in the expression level of GSK3β not protein kinase activity. It could correct the description in the paper.

Response 7: GSK3β related parts have been added or revised as follows.

In discussion :

  • Most of the previous studies (Leem et al., 2018; Wang et al., 2019) that reported changes in GSK3 β activation by exercise training applied protocols with high exercise intensity or relatively high exercise quantity, and Yang et al. reported that GSK3β showed little change at low strength, but it was reported to be significantly changed at medium and high strength. In addition, Pena et al. suggested that GSK3β hardly changed even when a relatively large amount of exercise was applied by applying a combination of aerobic and resistance exercise. In this study, in order to observe the interaction between exercise and lithium, it was considered that the change in exercise-induced GSK3β was not clearly seen when applied with low intensity from the viewpoint of minimizing exercise stimulation.

In abstract :

GSK3β, a neuroprotective factor in hippocampus in obese rats’

  • Lithium and low-intensity endurance exercise for 12 weeks increased the expression of BDNF, a neuroprotective factor in hippocampus in obese mice, and lithium treatment inhibited the activity of GSK3β.